# [Short] Graph Distance as Surprise: Free Energy Minimization in Knowledge Graph Reasoning

**Gaganpreet Jhajj**
SCIS
Athabasca University
gjhajj1@learn.athabascau.ca

**Fuhua Lin**
SCIS
Athabasca University
oscarl@athabascau.ca

## Abstract

In this work, we propose that reasoning in knowledge graph (KG) networks can be guided by surprise minimization. Entities that are close in graph distance will have lower surprise than those farther apart. This connects the Free Energy Principle (FEP) [1] from neuroscience to KG systems, where the KG serves as the agent's generative model. We formalize surprise using the shortest-path distance in directed graphs and provide a framework for KG-based agents. Graph distance appears in graph neural networks as message passing depth and in model-based reinforcement learning as world model trajectories. This work-in-progress study explores whether distance-based surprise can extend recent work showing that syntax minimizes surprise and free energy via tree structures [2].

## 1 Introduction

The Free Energy Principle (FEP) suggests that biological systems minimize surprise by maintaining accurate world models [1, 3, 4]. Recently, Murphy et al. [2] demonstrated that syntactic operations minimize surprise through shallow tree structures. They quantified surprise via tree depth (geometric complexity) and Kolmogorov complexity (algorithmic complexity), approximated through Lempel-Ziv compression [5, 6].

In FEP, agents minimize variational free energy $F = -\log P(o, s) - H[Q(s)]$, where $o$ represents observations, $s$ hidden states, $P$ the generative model, and $Q$ the agent's beliefs [1]. The first term, $-\log P(o, s)$, quantifies surprise: entities with high probability under the generative model (high $P(o, s)$) yield low surprise (low $-\log P(o, s)$). For syntactic trees, Murphy et al. [2] used tree depth to proxy this probability; we extend this principle to general graphs using shortest-path distance.

In active inference, minimizing free energy drives both perception (updating beliefs $Q(s)$) and action (selecting policies that reduce uncertainty) [3]. We apply this principle to KG reasoning: entities at shorter graph distances have a higher probability under the agent's graph-based generative model.

Knowledge graphs (KGs) are increasingly integrated with modern AI agents, with the ability to improve reasoning, memory, and planning [7, 8, 9, 10, 11, 12, 13, 14, 15, 16]. Unlike syntactic tree structures, KGs are directed graphs that can contain cycles and multiple paths between nodes (entities). In this preliminary work, we propose that surprise in KG reasoning corresponds to graph distance, where the KG serves as the agent's generative model. Entities that require shorter paths from context are unsurprising, whereas distant or disconnected entities are more surprising. This is unlike surprise-driven exploration in RL [17, 18], where agents maximize surprise to explore, FEP agents minimize surprise by maintaining accurate generative models. Our work connects the FEP to practical KG systems through shortest-path distance, providing theoretical foundations for graph neural networks [19, 20, 21] and model-based reinforcement learning [22, 23].

39th Conference on Neural Information Processing Systems (NeurIPS 2025) Workshop: .

## 2 From Syntax to Semantics

Murphy et al. [2] quantified syntactic surprise via tree depth. We extend this to arbitrary directed graphs with cycles. Given a KG $\mathcal{G} = (\mathcal{E}, \mathcal{R}, \mathcal{T})$ with entities $\mathcal{E}$, relations $\mathcal{R}$, and triples $\mathcal{T} \subseteq \mathcal{E} \times \mathcal{R} \times \mathcal{E}$, geometric surprise is:

$$S_{\text{geo}}(e \mid C) = \begin{cases} \min_{c \in C} d_{\mathcal{G}}(c, e) & \text{if path exists} \\ \alpha & \text{otherwise} \end{cases} \tag{1}$$

where $d_{\mathcal{G}}(c, e)$ is the shortest directed path length from context $c \in C$ to entity $e$ (computed via BFS, Appendix B), and $\alpha$ penalizes disconnection. Combined with algorithmic complexity [2]:

$$F(e \mid C) = S_{\text{geo}}(e \mid C) + \lambda K(\pi_{C \to e}) \tag{2}$$

where $K(\pi_{C \to e})$ is Kolmogorov complexity of the relation path, approximated via Lempel-Ziv compression, and $\lambda$ weights the components. For trees, this recovers Murphy's tree depth; for general graphs, it handles cycles naturally.

**Connection to FEP**: Under FEP, agents minimize $F = -\log P(o, s) - H[Q(s)]$ [1]. Interpreting the KG as the agent's generative model, we posit $-\log P(e \mid C) \propto d_{\mathcal{G}}(C, e)$: shorter distances indicate higher probability. Thus $S_{\text{geo}}$ implements the surprise term, while $K(\pi)$ approximates $H[Q(s)]$. Figure 1 illustrates this with a political KG example (detailed calculations in Appendix A).

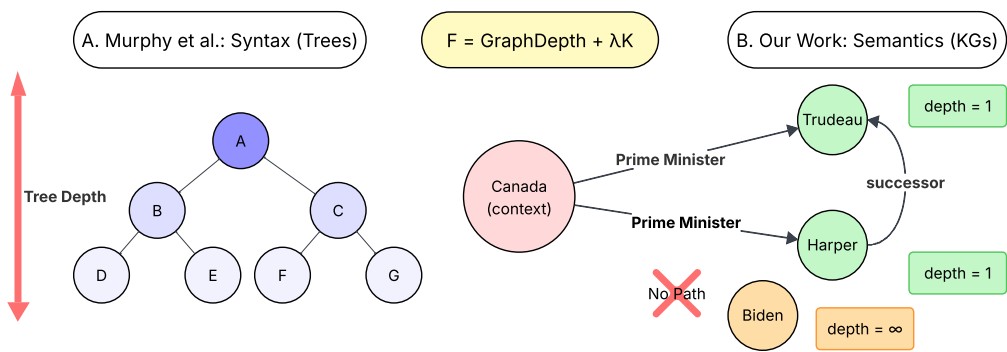

Figure 1: **From trees to graphs.** Murphy et al. [2] used tree depth for syntax; we use shortest path for semantic grounding. Given context "Canada", Canadian PMs (distance 1) have low surprise, Biden (distance $\infty$) has high surprise. Cycles handled naturally.

## 3 Theoretical Justification

Three principles justify shortest-path distance: **(1) Proper generalization**: For trees, recovers Murphy's tree depth exactly. **(2) Least-action**: Shortest paths minimize cumulative cost, aligning with active inference where agents minimize expected free energy [3]. **(3) Computational grounding**: In GNNs, $k$ message-passing iterations aggregate $k$-hop neighborhoods [19, 21]; minimizing iterations minimizes distance and surprise. Cycles pose no issue: FEP accommodates circular causality [24], and BFS handles cycles via visited sets (Appendix B).

## 4 Implications and Future Work

This work-in-progress connects FEP from neuroscience to KG reasoning in AI systems. The framework offers practical implications: (1) knowledge graph embeddings could preserve distance-based surprise structure [25]; (2) GNN depth could be selected to balance computational cost against accuracy; (3) KG-based agents implementing active inference could minimize expected free energy when selecting groundings.

Future work includes empirical validation on benchmark KG datasets (FB15k-237 [26], YAGO [27]), comparison to human semantic similarity judgments, integration with existing KG reasoning systems [10, 28, 9], and extension to temporal knowledge graphs [11].

## Acknowledgments

We acknowledge the support of the Natural Sciences and Engineering Research Council of Canada (NSERC), Alberta Innovates, Alberta Advanced Education, and Athabasca University, Canada.

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

# A Worked Example: Free Energy Calculations

We demonstrate free energy calculations using the Canadian Prime Minister knowledge graph from Figure 1.

## A.1 Scenario and Knowledge Graph

Consider query "Who is the Prime Minister?" with context $C = \{\text{Canada}\}$. The knowledge graph contains:

**Entities**: $\mathcal{E} = \{\text{Canada}, \text{Trudeau}, \text{Harper}, \text{Biden}\}$

**Relations**: (Canada, `pm`, Trudeau), (Canada, `pm`, Harper), (Trudeau, `successor`, Harper), (Harper, `predecessor`, Trudeau)

The successor/predecessor relations form a cycle: Trudeau $\leftrightarrow$ Harper. Importantly, Biden has no directed path from Canada (separate subgraph).

## A.2 Computing Geometric Surprise

Using BFS from Canada, we compute shortest directed paths:

- $d(\text{Canada}, \text{Trudeau}) = 1$ (direct via `pm`)
- $d(\text{Canada}, \text{Harper}) = 1$ (direct via `pm`)
- $d(\text{Canada}, \text{Biden}) = \infty$ (no path)

Therefore: $S_{\text{geo}}(\text{Trudeau}) = S_{\text{geo}}(\text{Harper}) = 1$, $S_{\text{geo}}(\text{Biden}) = \alpha = 5$.

The cycle between Trudeau and Harper does not affect distances: BFS selects the shortest path (direct edge) and handles cycles via visited set (Appendix B).

## A.3 Computing Algorithmic Complexity

For each grounding, we estimate Kolmogorov complexity via relation path patterns:

**Trudeau & Harper**: Path $\pi = [\text{pm}]$ is a frequent relation in political KGs, yielding high compression (low $K(\pi)$).

**Biden**: No path from Canada. The grounding requires irregular cross-country reasoning not represented in the graph (high $K(\pi)$).

## A.4 Free Energy Results

Combining components with $\lambda = 1$:

| Entity | $S_{\text{geo}}$ | $K(\pi)$ | $F$ |
|--------|--------|--------|--------|
| Trudeau | 1 | Low | ~1.3 |
| Harper | 1 | Low | ~1.3 |
| Biden | 5 | High | ~5.5 |

**Interpretation**: Common groundings (Trudeau, Harper) exhibit low free energy: (1) short distance (1 hop), (2) regular relation patterns. The rare grounding (Biden) exhibits high free energy: (1) disconnection (no path), (2) irregular pattern. The framework correctly identifies both Trudeau and Harper as plausible (both were Canadian PMs) while rejecting Biden (US president).

This demonstrates three key properties: **(1)** cycles handled naturally, **(2)** multiple valid answers coexist with equal surprise, **(3)** disconnected entities correctly penalized.

# B Mathematical Details

## B.1 Breadth-First Search Algorithm

Given directed graph $\mathcal{G} = (\mathcal{E}, \mathcal{R}, \mathcal{T})$ and context $C \subseteq \mathcal{E}$, we compute $S_{\text{geo}}(e \mid C)$ via BFS:

---

**Algorithm 1** Compute Geometric Surprise

---

**Require:** Knowledge graph $\mathcal{G}$, context $C$, target entity $e$
**Ensure:** Geometric surprise $S_{\text{geo}}(e \mid C)$
1: Initialize: $d(c) \leftarrow 0$ for all $c \in C$; $d(v) \leftarrow \infty$ for $v \notin C$
2: $Q \leftarrow C$ (queue), $V \leftarrow C$ (visited set)
3: **while** $Q \neq \emptyset$ **do**
4:    $u \leftarrow$ dequeue from $Q$
5:    **for** each outgoing edge $(u, r, v) \in \mathcal{T}$ **do**
6:      **if** $v \notin V$ **then**
7:        $d(v) \leftarrow d(u) + 1$
8:        $V \leftarrow V \cup \{v\}$, enqueue $v$ to $Q$
9:      **end if**
10:    **end for**
11: **end while**
12: **return** $d(e)$ if $d(e) < \infty$, else $\alpha$

---

**Properties**: **(1)** *Correctness*: BFS finds shortest paths in $O(|\mathcal{E}| + |\mathcal{T}|)$ time. **(2)** *Cycle handling*: Visited set $V$ prevents re-visiting nodes, ensuring termination. **(3)** *Directionality*: Only outgoing edges followed, respecting direction.

## B.2 Kolmogorov Complexity Approximation

We approximate $K(\pi_{C \to e})$ via Lempel-Ziv compression: **(1)** Extract relation sequence $\pi = [r_1, \ldots, r_k]$ from shortest path. **(2)** Encode as string (e.g., "pm|successor"). **(3)** Compress with LZ77. **(4)** Compute ratio $K(\pi) = $ compressed/original.

**Interpretation**: Regular patterns (frequent relations, short sequences) achieve high compression (low $K$). Irregular patterns (rare relations, long sequences) achieve low compression (high $K$). This approximates Kolmogorov complexity, which is uncomputable [5]. Murphy et al. [2] use the same approximation for syntactic patterns.

## B.3 Connection to Active Inference

In active inference, agents minimize expected free energy $G(\pi)$ [3, 4]:

$$G(\pi) = \underbrace{D_{KL}[Q(o|\pi)\|P(o)]}_{\text{Pragmatic}} + \underbrace{\mathbb{E}_{Q(o|\pi)}[H[P(s|o)]]}_{\text{Epistemic}} \tag{3}$$

balancing pragmatic value (exploitation) and epistemic value (exploration).

**Pragmatic value**: Entities at shorter distances are more likely: $P(\text{observe } e \mid C)$ increases as $S_{\text{geo}}$ decreases, making low-distance entities preferred for goal-directed actions.

**Epistemic value**: Entities at longer distances provide higher information gain: observing distant entities reduces uncertainty about unexplored graph regions, making high-distance entities preferred for exploration.

Our $S_{\text{geo}}$ implements pragmatic value: low surprise entities preferred for exploitation. Extensions could weight distance inversely for epistemic value, valuing high-surprise entities for exploration.

