# OpenReview forum: "Graph Distance as Surprise: Free Energy Minimization in Knowledge Graph Reasoning"
_NeurIPS.cc/2025/Workshop_Mexico_City/NORA — NeurIPS 2025 Workshop NORA Poster_

### Official Review · Reviewer_WnMH · 2025-11-01
**An idea article**

**Rating:** 4
**Confidence:** 4

**Review:**

This manuscript proposes an idea without any experiments. The idea sounds very interesting. However, the provided theoretical justification is more like arguments rather than rigorous proof. In addition without experimental validation, it is hard to judge.

---

### Official Review · Reviewer_zeSY · 2025-11-03

**Rating:** 5
**Confidence:** 4

**Review:**

This paper focuses on knowledge graph reasoning, which aims to predict the truth values of unseen facts in a knowledge graph. Specifically, the proposed method is inspired by the Free Energy Principle. The basic idea is to minimize the suprise during reasoning. This work is still in progress, therefore, there is no experimental results.

The strengths include:
1. a novel idea derived from neuroscience.
2. some interesting theoretical analyses.

The weaknesses include:
1. The proposed method seems to be purely distance-based, which means the semantic information (e.g., entity and relation names) is not utilized.
2. Although the title contains term "knowledge graph reasoning", the problem statement is not clearly described in the content. Is that the traditional link prediction task setting?
3. Missing experimental results. However, considering this is a two-page short paper, this is not a fatal problem.

---

### Official Review · Reviewer_c7Uw · 2025-11-03
**Promising work that needs more depth**

**Rating:** 5
**Confidence:** 3

**Review:**

Summary:
The work shows promise and innovation by integrating the Free Energy Principle (FEP) from cognitive science into Knowledge Graph (KG) reasoning. While the idea is promising and the mathematics enticing, it falls a bit short and fails to hold up with a small example only to demonstrate its validity.

Strengths:
- Novelty: the idea is original and to the reviewer's knowledge, yet to be tested or applied in other works in the domain
- Application: the idea applies to a hot domain of interest in Artificial Intelligence (AI) reasoning

Weaknesses:
- Section A.2: it is hard (and unclear) why alpha is equal to the value 5 for the entity Biden. Alpha is said to be infinite when no path exists, but the choice of assigning the value 5 seems unjustified and unexplained.
- Section A: the worked example is insufficient to suggest this is a method that can scale or be applied to other KGs. In addition, the example itself is small and limited to one domain (politics) and a handful of entities and relations.
- Figure 1: The method does not justify why it would be a more appropriate choice than BFS or any other graph traversal algorithms. In addition, while it does make the case for conserving cycles, it does not in any way seem to make use of these relations (e.g., is a successor Prime Minister the better likely answer to a current query involving the Prime Minister?)

---

### Official Review · Reviewer_wE6z · 2025-11-06
**Review for submission #2**

**Rating:** 4
**Confidence:** 4

**Review:**

This paper proposes using graph distance to quantify surprise in the Free Energy Principle (FEP) for knowledge graph reasoning. While the core idea has some interest, the paper suffers from fundamental issues.

The paper demonstrates a misunderstanding of knowledge graph modeling. In their primary example (Figure 1 and Appendix A), "Prime Minister" is treated as a relation (edge) connecting Canada to Trudeau. However, "Prime Minister" is a role or position—a concrete entity that should be represented as a node in the graph. In standard knowledge graph design (as seen in Wikidata, DBpedia, etc.), this would be modeled as Canada having a "hasLeader" relation to Trudeau, with Trudeau holding the position of Prime Minister through a "holdsPosition" relation.

The theoretical justification for mapping FEP concepts to graph distance is weak. The paper asserts that shortest-path distance corresponds to surprise but does not rigorously justify why this particular metric best captures the FEP framework among many possible graph measures.

Using graph distance for knowledge graph reasoning is well-established. The connection to FEP feels superficial. It is more of a reframing than a genuine theoretical advance.

---

### Official Review · Reviewer_LALx · 2025-11-06
**Traversing knowledge graphs by minimizing surprisal**

**Rating:** 7
**Confidence:** 2

**Review:**

**Summary**

The paper proposes a formalism of KG traversal through the lens of the free energy principle. This is operationalized this by linking graph distance to surprise with closer entities in a KG being less surprising and distant or disconnected entities being more surprising. The proposed traversal combines surprisal with algorithmic complexity as a means to measure how "regular" a path is between entities. Together, distance and complexity provide a *free energy* interpretation of traversing knowledge graphs where short and simple connections have low surprise, while long or irregular ones are more surprising. This extends prior work on syntactic tree depth in language, which stands as an interesting scientific extension. Conceptually, the bridge between the free energy principle and AI’s graph reasoning is novel. The paper is framed strictly theoretical but that may help unify how we the community perceives knowledge graph embeddings, GNNs, and exploration in reasoning agents.

**Feedback**

I thoroughly enjoyed reading this paper and it certainly inspired some ideas on my end, however, I find that grounding the formalism better, would improve the manuscript. Generally, this boils down to two points;
- While the paper positions itself theoretically, it would benefit from being clearer on how geometric surprise and algorithmic complexity interact. The relationship between the two terms is unclear and there's no clear sense of when or how one should dominate the other. The provided example, is clear but somewhat too trivial to illustrate the model’s nuance. Because the geometric distance term completely dictates the result (connected vs. disconnected), the complexity term doesn’t meaningfully change the interpretation. It might be good to include a case where the geometric distance is identical for two candidates but their relation paths differ in regularity, or long paths with regular complexity. This might make the value of the algorithmic complexity term more tangible and the formalism more convincing.
- The paper would benefit from clearer grounding on how the proposed formalism might be operationalized. Currently it is difficult to envision how an agent or LLM-KG system would compute or approximate algorithmic complexity during reasoning. Providing even speculative examples of how this might work, e.g. whether compression is derived from textual encoding, embedding similarity, or model perplexity would make the contribution more concrete and aid the reader in understanding what the formalism intends to address. Does it aim to reduce reasoning uncertainty, improve retrieval grounding, or offer a principled way to rank or filter KG relations to counter ungrounded but probable generations? How would this perspective change the behavior of an agent using a KG compared to existing heuristics? A clear statement of what problem this approach helps solve, and where it would strengthen the research direction and would make the paper’s contribution feel both more targeted and more impactful.

---

### Official Review · Reviewer_6sc4 · 2025-11-07
**While the paper introduces a creative and potentially impactful idea, it remains speculative and underdeveloped. The absence of empirical evaluation, incomplete theoretical motivation, and lack of clarity in some formulations make it unsuitable for acceptance in its current form.**

**Rating:** 3
**Confidence:** 4

**Review:**

Summary:
This paper proposes an intriguing conceptual link between the Free Energy Principle (FEP) from neuroscience and reasoning in knowledge graphs (KGs). The authors argue that entities closer in a KG should yield lower “surprise,” extending prior work that linked syntactic tree depth to surprise minimization. They formalize this idea using shortest-path distance and introduce a free-energy formulation combining geometric distance with Kolmogorov complexity of relation paths. A small illustrative example involving political entities is provided, and the authors discuss potential implications for graph neural networks, KG embeddings, and active inference systems.

Strengths:
The paper presents a creative and elegant theoretical idea that extends FEP-inspired reasoning from syntax to general graph structures. The notion of using graph distance as a proxy for surprise is novel and conceptually coherent, offering an interesting perspective that connects cognitive theory with graph-based AI reasoning. The presentation is mostly clear and includes a simple worked example that helps illustrate the intuition behind the proposal.

Weaknesses:
The paper lacks any empirical validation to support its claims. No experiments are provided to test whether graph distance meaningfully reflects semantic plausibility or reasoning performance on real-world KGs. As a result, the contribution remains entirely conceptual. The work also feels premature, with several proposed extensions (e.g., integration with embeddings or GNNs) left undeveloped. Moreover, the sudden inclusion of Kolmogorov complexity is insufficiently justified and disrupts the theoretical coherence of the framework. At times, the writing becomes confusing, mixing FEP terminology with graph concepts without consistent grounding.

---

### Official Review · Reviewer_8f25 · 2025-11-07
**Theoretical framework needs empirical grounding**

**Rating:** 7
**Confidence:** 3

**Review:**

Strengths:

The formalization is clean. Equation 1 defines geometric surprise as shortest-path distance, handling cycles through BFS. This naturally extends tree depth to general graphs. The worked example with Canadian PMs versus Biden effectively demonstrates the concept. The connection to GNN message passing depth is insightful.

Major weaknesses:

No empirical validation whatsoever. The authors mention FB15k-237 and YAGO but present zero results. Without data, we cannot assess whether distance actually correlates with semantic surprise or human judgments.

The Kolmogorov complexity term is handwaved. Appendix B.2 mentions Lempel-Ziv compression but provides no concrete algorithm, parameters, or examples. The worked example just labels things "low" and "high" without calculation.

The disconnection penalty α=5 appears arbitrary. No justification or sensitivity analysis.

The additive combination F  lacks theoretical motivation. Why addition? Why not other functional forms?